# What is the quantity, quality and type of systematic review evidence available to inform the optimal prescribing of statins and antihypertensives? A systematic umbrella review and evidence and gap map

Liz Shaw ,[1] Simon Briscoe,[1] Michael P Nunns,[1] Hassanat Mojirola Lawal ,[1] G J Melendez-Torres,[1] Malcolm Turner,[1,2] Ruth Garside,[1,3] Jo Thompson Coon [1]

For numbered affiliations see end of article.

**Correspondence to**
Dr Liz Shaw;
E.H.Shaw@exeter.ac.uk

## ABSTRACT

**Objectives** We aimed to map the systematic review evidence available to inform the optimal prescribing of statins and antihypertensive medication.

**Design** Systematic umbrella review and evidence and gap map (EGM).

**Data sources** Eight bibliographic databases (Cochrane Database of Systematic Reviews, CINAHL, EMBASE, Health Management Information Consortium, MEDLINE ALL, PsycINFO, Conference Proceedings Citation Index—Science and Science Citation Index) were searched from 2010 to 11 August 2020. Update searches conducted in MEDLINE ALL 2 August 2022. We searched relevant websites and conducted backwards citation chasing.

**Eligibility criteria for selecting studies** We sought systematic reviews of quantitative or qualitative research where adults 16 years+ were currently receiving, or being considered for, a prescription of statin or antihypertensive medication. Eligibility criteria were applied to the title and abstract and full text of each article independently by two reviewers.

**Data extraction and synthesis** Quality appraisal was completed by one reviewer and checked by a second. Review characteristics were tabulated and incorporated into an EGM based on a patient care pathway. Patients with lived experience provided feedback on our research questions and EGM.

**Results** Eighty reviews were included within the EGM. The highest quantity of evidence focused on evaluating interventions to promote patient adherence to antihypertensive medication. Key gaps included a lack of reviews synthesising evidence on experiences of specific interventions to promote patient adherence or improve prescribing practice. The evidence was predominantly of low quality, limiting confidence in the findings from individual reviews.

**Conclusions** This EGM provides an interactive, accessible format for policy developers, service commissioners and clinicians to view the systematic review evidence available relevant to optimising the prescribing of statin and antihypertensive medication. To address

## STRENGTHS AND LIMITATIONS OF THIS STUDY

⇒ Our evidence and gap map (EGM) represents the results of a comprehensive, systematic search for systematic review evidence in this field.
⇒ The EGM includes only systematic review evidence, and thus, there may be primary research relevant to the topic which may sit within the identified gaps.
⇒ Our search strategy did not capture systematic reviews, which may have met our eligibility criteria that were used to inform National Institute for Health and Care Excellence (NICE) guidelines relevant to the review topic. These reviews were not retrieved by our bibliographic database searches, thus highlighting issues regarding the lack of visibility of the reviews contributing to NICE guidelines.

high-quality research, future reviews should be conducted and reported according to existing guidelines and address the evidence gaps identified above.

## INTRODUCTION

With an estimated 17.9 million deaths during 2019, cardiovascular disease (CVD) is the leading cause of death worldwide.[1] One study estimates the total costs of heart failure for 98.7% of the world's population to be US$108 billion,[2] with individuals from black, Asian and other minority ethnic backgrounds, low socioeconomic status or those living with diabetes or severe mental illness being more at risk of the disease.[3] Along with earlier detection of risk factors for CVD and promotion of lifestyle changes, high-intensity statins are recommended over the use of low-intensity statins as an effective treatment to prevent CVD.[4] While a key target set by the World Health Organisation (WHO) that a

minimum 50% of eligible individuals will be in receipt of statins has been met in high-income countries,[5] the prescribing of this type of medication in the UK has not yet reached optimal levels.[4]

Hypertension is another risk factor in the development of CVD[6] and can be controlled through use of antihypertensive medication, sometimes in combination with statins, to reduce the occurrence of CVD.[7] While, the rate of hypertension remained largely unchanged between 1990 and 2019, the burden of illness shifted from high to middle-income and low-income countries.[8] There are early signs that awareness and treatment of hypertension within some high-income countries, such as the UK, has decreased within the last 5 years.[9 10] Patient non-adherence is also an issue of concern, with adherence decreasing with higher numbers of antihypertensive medications prescribed per patient.[11]

To achieve optimal prescribing of statins and antihypertensive medication, it is essential that the factors influencing both their prescription and patient adherence are fully understood. There is already a diverse range of existing systematic review evidence in this area which could be used by policy-makers, commissioners and clinicians to design and deliver services or research to support the needs of those living with, or at risk of, CVD. To fully understand and make use of this existing evidence base, this paper focuses on the creation of an evidence and gap map (EGM) to display the quantity, quality and type of quantitative and qualitative systematic review evidence available to inform the optimal prescribing of statins and antihypertensives.

We were interested in systematic reviews which synthesised the following:

► Evidence regarding the effectiveness or experiences of interventions intended to improve prescribing practices or patient adherence.
► Evidence on the effectiveness or experiences of interventions intended to improve implementation of interventions intended to improve prescribing practices or patient adherence.
► Evidence focusing on practitioner views or perceptions of making prescribing decisions.
► Guidelines intended to inform prescribing practice.

This paper is part of a broader systematic mapping review commissioned by National Health Service (NHS) England and NHS Improvement (NHSE-I) relating to the optimising of the prescribing of three different groups of medication: medications to treat CVD, drugs that can cause dependency (DCD; including opioids, benzodiazepines, gabapentinoids and non-benzodiazepine hypnotics) and antidepressants.[12]

## METHODS

The methods used for the searching, screening, data extraction and quality appraisal processes of the evidence included in this EGM were in accordance with the University of York's Centre for Reviews and Dissemination

guidelines.[13] We also consulted guidance for the development of EGMs.[14] Our methods were outlined in our preregistered review protocol and are summarised below according to the Preferred Reporting Items for Systematic Reviews and Meta-Analysis (PRISMA) reporting guidance.[15 16]

### Search strategy

The first stage of our systematic umbrella review involved conducting searches of bibliographic databases and websites for relevant literature. Our bibliographic database search strategy combined terms for optimising prescribing with terms for statins and antihypertensives and CVD. The search strategy was developed using MEDLINE (via Ovid) using a combination of free-text terms and controlled headings. Search terms were derived from preidentified relevant papers and inspected by stakeholders with expertise in optimising prescribing. Searches were run on 11 August 2020 in the following bibliographic databases: Cochrane Database of Systematic Reviews (via Cochrane Library), CINAHL (via EBSCO), EMBASE (via Ovid), Health Management Information Consortium (via Ovid), MEDLINE ALL (via Ovid), PsycINFO (via Ovid), Conference Proceedings Citation Index—Science and Science Citation Index (both via Web of Science, Clarivate Analytics). We also searched systematic review database Epistemonikos and preprint server medRxiv. This selection of databases reflects our desire to comprehensively search for systematic review evidence within this field and the broad scope of an umbrella review. The databases selected provide coverage of a variety of healthcare disciplines relevant to our research question.

An historical date limit of 2010 was applied to all search results to prioritise the most recent systematic reviews, evaluating interventions and experiences most relevant to current medical settings, for inclusion in the EGM. We carried out backwards citation chasing for all reviews which met our inclusion criteria, searched topically relevant websites and pursued full texts for conference abstracts and review protocols identified through our searches. We also contacted review authors where we could not access full texts. Update searches were conducted in MEDLINE ALL (via OVID) on 2 August 2022 and 23 June 2023. We limited our update search to MEDLINE since all the studies identified through our original search were indexed in this database. Full details of database search strategies are reproduced in online supplemental appendix A (which also includes search terms for DCD, which was a second topic that our client asked us to investigate in our mapping review. The findings associated with this second topic are available elsewhere).[12]

### Study selection

After running our search strategy, the second stage of our umbrella review involved two reviewers (LS2, MPN or HML) screening a sample of 100 titles and abstracts to ascertain the utility of our inclusion criteria for this review.

**Table 1** Review inclusion criteria

| PICO/PICo criteria | Inclusion criteria |
|---|---|
| Population | Mean age ≥16<br>Receiving/being considered for a prescription for statins and/or antihypertensives. |
| Intervention/phenomenon of interest | For systematic reviews of quantitative evidence, interventions which aimed to improve one or more of the following:<br>► Adherence to prescribed medication of interest.<br>► Discontinuation of medications of interest.<br>► Prescriber adherence to clinical guidance for prescribing.<br>► Prescriber practices.<br>► Implementation of an intervention to enhance patient adherence or prescriber practices.<br>Interventions could be conducted at the level of system or that of the patient/prescriber.<br>For systematic reviews of qualitative evidence, review foci could include the views or experiences of patients, carers or clinicians for:<br>► Healthcare consultations to discuss initiation, reviewing or discontinuing a prescription.<br>► Interventions to improve adherence*/prescribing practice.<br>► Reasons for adherence or non-adherence to prescribed medication.<br>► Making prescribing decisions. |
| Comparator/context | Any |
| Outcome | Relevant to the aims of our review and stated clearly in the study abstract.<br>For reviews of quantitative evidence, examples may include:<br>► Measures of patient adherence.<br>► Measures of prescriber adherence to prescribing guidelines.<br>► Measures of success of implementing interventions to promote adherence, prescribing or discontinuation.<br>For reviews of qualitative evidence, please see phenomenon of interest section above. |
| Study design | Systematic reviews which met the following criteria:<br>► Clearly stated research question.<br>► Indicate which sources were searched.<br>► Reproducible search strategy and search date.<br>► Clearly defined inclusion and exclusion criteria and study selection methods.<br>► Critically appraise and report the quality of included studies.<br>► Reproducible data synthesis strategy.<br>Encompasses: systematic reviews of quantitative and qualitative literature, systematic reviews of reviews, systematic reviews of guidelines, scoping and rapid reviews. |
| Date limit | Systematic reviews published before 2010 were excluded. This was to prioritise the most recent systematic reviews, evaluating interventions and experiences most relevant to current medical settings, for inclusion in the EGM. |

*For convenience, we use the term 'patient adherence' throughout this report to encapsulate this continual decision-making process in which the patient can choose, and/or be enabled, to play an active role when taking their medication as prescribed.
EGM, evidence and gap map.

The revised inclusion criteria (table 1) were then applied independently by two reviewers to each citation retrieved via our search strategy (LS, MPN), with disagreements resolved through discussion or referral to a third reviewer (SB). Eligibility of full texts was assessed using the same method. Screening decisions were recorded in EndNote V.X8 software (Clarivate Analytics, Philadelphia, Pennsylvania, USA) and the study selection process was detailed within a PRISMA-style flow chart.

### Data extraction and quality appraisal
After title and abstract and full-text screening of the articles identified via our search strategy, two reviewers carried out a pilot of our data extraction form on five reviews. The finalised data extraction and quality appraisal form was applied to all included systematic reviews by one reviewer (MPN and LS) and checked by a second (SB, LS, HML and MPN), supported by EPPI-Reviewer (V.4.11.5.2) software. Disagreements were resolved through discussion or referral to a third person.

We then conducted quality appraisal in two stages. Reviews were first appraised using four modified criteria from the Collaboration for Environmental Evidence Synthesis Assessment Tool (CEESAT)[17]:

1. Search strategy: Is the approach to searching clearly defined, systematic and transparent?
2. Is the search comprehensive?
3. Does the review critically appraise each study?
4. During critical appraisal, was an effort made to minimise subjectivity?

Only systematic reviews which met all four criteria were prioritised for full quality appraisal using the AMSTAR-2 (Assessing the Methodological Quality of Systematic review, version 2) guidance,[18] modified to incorporate reporting standards for qualitative evidence synthesis.[19] Reviews appraised using only the CEESAT criteria were given an overall quality rating of 'critically low'. This process of quality appraisal represents a deviation from our protocol and was necessary due to the high number

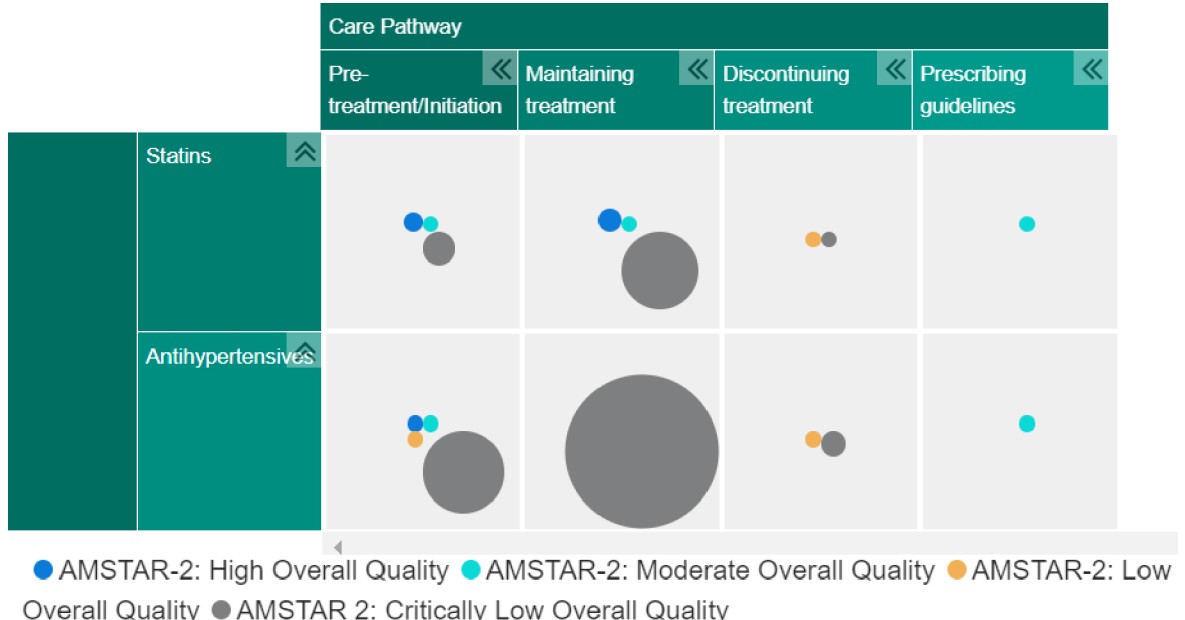

**Figure 1** Evidence and gap map. AMSTAR-2 = Assessing the Methodological Quality of Sysematic Reviews - version 2

of studies eligible for inclusion in the EGM. Thus, we decided to use the process described above to priories the inclusion of the most robust evidence for inclusion in the EGM.

Full details of the data extracted from each review, including CEESAT and AMSTAR-2 criteria, are provided elsewhere.[12]

### Data analysis and presentation

In the final stage of our umbrella review, details of the systematic reviews prioritised for inclusion in the EGM were summarised in a table and described narratively. We used EPPI-Mapper software to structure the EGM according to medications of interest and patient care pathway.[20]

The initial care pathway was based on NICE guidelines for statins and antihypertensives, optimising prescribing and shared decision-making.[5 21–23] To ensure the pathways reflected both recommended practice and implementation within 'real-life' settings, we also drew on a systematic review exploring patient experiences of medication taking and consulted with patients and carers with experience of seeking, being prescribed, taking and/or discontinuing statins and/or antihypertensives.[24] We also asked clinicians if the pathway we developed reflected their clinical experience working within the area. Finally, we produced a condensed version of the map which focused on pretreatment/initiation, maintenance, discontinuation and development of guidelines.

The map presents the systematic review evidence according to the 'overall quality' rating provided by the AMSTAR-2. The colour and size of the bubble indicate the quality rating and number of systematic reviews for that medication type at that part of the care pathway, respectively. Systematic reviews relevant to more than one type of medication or position on the

care pathway are represented more than once within the map. Filters also allow the map user to alter the type of evidence shown in the map. Policy, clinical and patient stakeholders were asked to comment on the presentation and content of the final EGM to ensure findings were accessible to our intended audiences.

### Patient and public involvement

We worked alongside a group of stakeholders with a variety of expertise in the topic area throughout the production of this EGM. Stakeholders included representatives from NHSE-I, NICE, clinicians and people with experience of taking one or more of the medications of interest. Key stages at which all stakeholders, including patients and members of the public, contributed during the project included; development of the protocol, including providing feedback on the appropriateness of review questions, supporting development of the bibliographic database search strategies and list of relevant topic websites to be searched. Clinical stakeholders and patients/members of the public also provided insight on the list of included studies, including resolving queries regarding eligibility for inclusion, supported the development of the patient care pathway and also provided feedback on the presentation of the EGM and gaps in systematic review evidence within the EGM and how findings should be disseminated.

### RESULTS

Following deduplication, 1694 unique records were found through our bibliographic database and supplementary searches. 1516 of these were excluded during title and abstract screening, with a further 113 excluded after the reviewing of full texts. Fourteen systematic reviews were

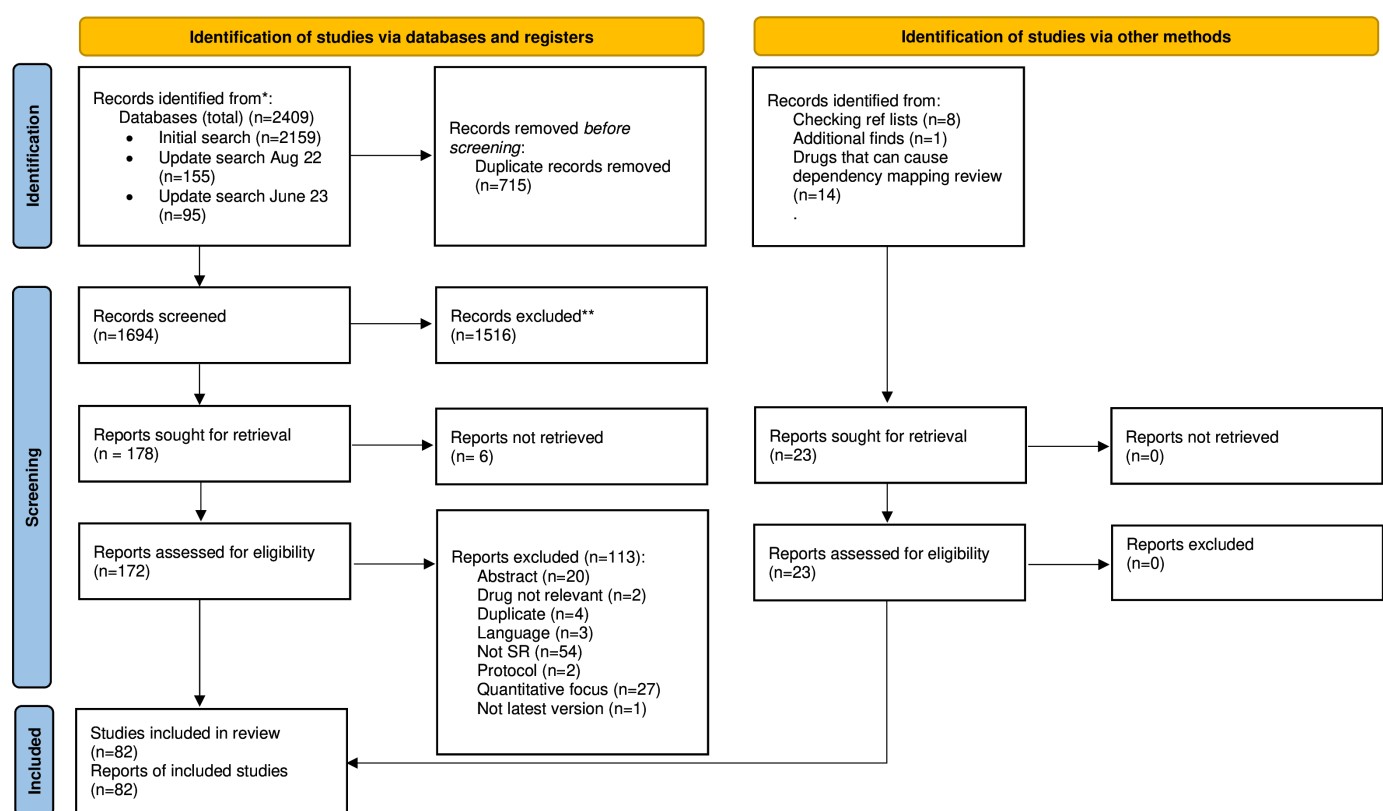

**Figure 2** PRISMA diagram. PRISMA 2020 flow diagram for new systematic reviews which included searches of databases, registers and other sources. *Consider, if feasible to do so, reporting the number of records identified from each database or register searched (rather than the total number across all databases/registers). **If automation tools were used, indicate how many records were excluded by a human and how many were excluded by automation tools. Page *et al.*[107] PRISMA=Preferred Reporting Items for Systematic Reviews and Meta-Analysis

identified through the search strategy intended to identify other medication groups as part of a wider piece of work (ref withheld for peer review). In total, 80 systematic reviews were eligible for inclusion in the review. These studies are presented within our EGM(access here, https://eppi.ioe.ac.uk/CMS/Portals/35/EGM_University_of_Exeter_August_2021.html also shown in figure 1). Figure 2 summarises the study selection process and reasons for exclusion at full text, with a list of full reasons for exclusion provided in online supplemental appendix B.

### Characteristics of evidence included within umbrella review
Medications of interest within the 80 reviews included in our umbrella review were: statins (n=10),[25–34] antihypertensives (n=38),[35–72] and statins and antihypertensives (n=34).[73–104] Nine of the reviews synthesised qualitative evidence,[27 31 42 50 55 58 60 61 83 104] four of which also included quantitative evidence in a mixed evidence synthesis.[42 50 58 61] Of the reviews synthesising quantitative evidence evaluating interventions, interventions focused on: enhancing patient adherence to medication as prescribed (n=57),[25 30 32–35 37–41 43–46 48 49 51 53 54 56 57 59 62 63 65–71 73–81 86 87 89–96 98–103 105] deprescribing a prescribed medication (n=6)[36 46 72 80 88 93] or optimising the prescribing of medication to meet the needs of the patient (n=13).[26 28 29 38 52 64 71 84 85 87 90 100 106] Three of these reviews were judged by reviewers to have

dual aims.[71 87 90] An overview of the characteristics of all reviews included in the EGM is included in online supplemental appendix C.

### Characteristics of evidence included in EGM
Twenty-two of the 80 systematic reviews included in our umbrella review were fully appraised using the modified AMSTAR-2 tool. Ten were 'high' quality,[27 28 54 72 73 81 98–100 103] five were of 'moderate' quality,[32 53 65 84 95] four of 'low' quality[26 57 64 82] and three 'critically low' quality.[44 63 96] The remaining reviews scored poorly on one or more of the CEESAT criteria during the first stage of quality appraisal, and were thus awarded a 'critically low' overall quality rating. The full quality appraisal ratings for each review can be found in online supplemental appendices D and E.

Table 2 provides a summary of the systematic review evidence included in the EGM.

### Summary of evidence on the patient care pathway
Figure 3 presents a visual summary of the 80 reviews included in the EGM, presented according to the volume of evidence available which corresponds to the different aims of interest to this umbrella review. Further detail is summarised within "Summary of evidence and gaps" below.

**Table 2** Summary of included reviews

| Topic/type of evaluation | Type of evidence | No of reviews* | Appraised using AMSTAR-2 Y/N: N (overall quality rating) | Medication of interest |
|---|---|---|---|---|
| Prescriber, patient and/or family/carer views of issues relating to prescribing and/or adherence to medications of interest | Qualitative | 10† | Y: 1 (High: 1) | Statins: 1 |
| | | | N: 9 | AH:6 Statins: 1 Statins+AH: 2 |
| Evaluating intervention to optimise prescribing | Quantitative | 15 | Y: 6 (high4:, moderate:1, low: 2) | AH: 2 Statins: 2 Statins+AH: 2 |
| | | | N: 9 | AH:3 Statins: 1 Statins+AH: 5 |
| Evaluating intervention to deprescribe medication | Quantitative | 5 | Y: 1 (critically low: 1) | Statins+AH: 1 |
| | | | N: 4 | AH:2 Statins+AH: 2 |
| Evaluating intervention to optimise patient adherence to a medication | Quantitative | 57 | Y: 14 (high: 7, moderate: 3, low: 1, critically low: 3) | AH: 8 Statins: 1 Statins+AH: 5 |
| | | | N: 43 | AH: 20 Statins: 4 Statins+AH: 19 |

AMSTAR-2: Assessing the Methodological Quality of Systematic Reviews
*n>80 as some reviews were coded as having multiple aims.
†Four reviews also included quantitative evidence.
AH, antihypertensive; N, no; Y, yes.

## Summary of evidence and gaps

Figure 3 illustrates a large cluster of evidence relating to the evaluation of interventions to promote maintaining adherence to medication, particularly to treat hypertension. The next biggest cluster of evidence is relevant to the 'pretreatment' or 'initiation' stage of the pathway with antihypertensive medication.

However, overall the quality of the evidence at these segments of the care pathway was predominantly of critically low quality.

The quantity of qualitative evidence available is low and, while the evidence examines the perceptions of patients, carers and/or clinicians of issues affecting medication prescription or adherence, none of the systematic reviews

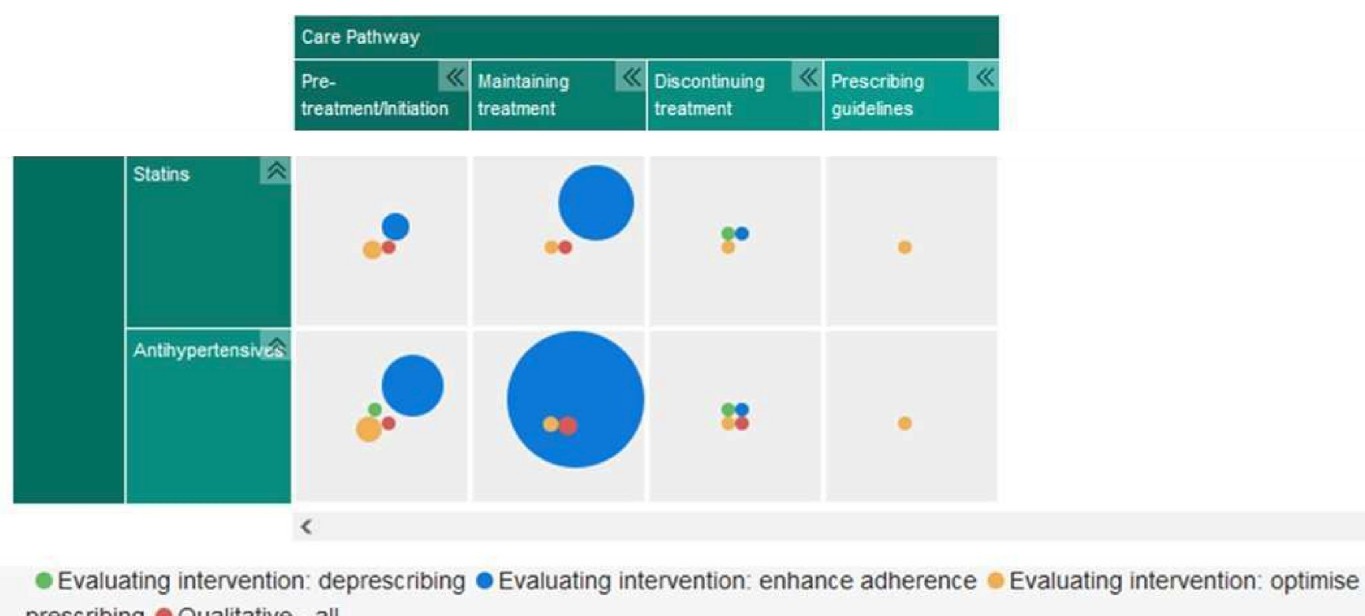

● Evaluating intervention: deprescribing ● Evaluating intervention: enhance adherence ● Evaluating intervention: optimise prescribing ● Qualitative - all

**Figure 3** Focus/aims of systematic review evidence included in the map.

explore experiences of specific interventions intended to optimise prescribing in some way. Only one of the systematic reviews of qualitative evidence was of 'high' quality.

### Pretreatment/initiation

Five reviews synthesised qualitative research.[27 31 50 55 104] One high-quality review of qualitative evidence explored patient and family members views of the barriers and enablers to treatment adherence for heterozygous familial hypercholesterolaemia.[27] Three reviews of 'critically low' quality review synthesised qualitative evidence relevant to the prescribing of antihypertensives.[31 50 55] Two of these reviews included a range of different perspectives,[31 50] while the other focused on patient perspectives.[55] One 'critically low' quality review explored practitioner and patient views of prescribing or adhering to statin and/or antihypertensive medication.[104]

Thirty-two systematic reviews synthesised quantitative evidence to promote medication adherence. Statins were the medication of interest for 10 reviews, 2 of 'high' overall quality[81 103] and 8 of 'critically low' overall quality.[33 75 87 91 93 94 101 102] Antihypertensives were the medication of interest in 20 systematic reviews; 3 were of 'high' quality,[54 81 103] 2 of 'moderate' overall quality[53 65] and 15 were 'critically low' overall quality.[37 43 45 49 51 56 66 71 75 87 91 94 95 101 102] Ten reviews sat across both statin and antihypertensive care pathways.[74 79 80 85 88 90 91 97 98 105]

Six systematic reviews synthesised quantitative evidence evaluating interventions to optimise prescribing, where the medication of interest was statins; 'high' quality (n=2),[28 100] 'moderate' quality (n=1)[84] and 'critically low' quality (n=3).[29 85 87] One 'critically low' quality review synthesised quantitative evidence evaluating interventions to reduce the prescribing of fall risk increasing drugs in older adults.[47] The medication of interest within one 'critically low' quality review was both statins and antihypertensives,[106] and antihypertensives alone in a high-quality review.[72]

### Maintaining treatment

Two high quality,[27 72] and seven 'critically low' quality,[31 33 50 55 80 105 106] reviews of qualitative evidence included in the 'pretreatment/initiation' part of pathway, were also relevant here.

In addition, one 'critically low' quality review explored reasons associated with adherence or non-adherence to statin medication,[61] and a further three 'critically low' quality reviews combined quantitative and qualitative data to explore factors relating to adherence/non-adherence where antihypertensives and/or statins were the medication of interest.[16 24 52]

Twenty-five systematic reviews synthesised evidence evaluating interventions to improve adherence to statins. Five were of 'high' overall quality,[2 13 54 58 71] one of 'moderate' overall quality,[51] and 19 of 'critically low' overall quality. In total, 49 systematic reviews synthesised quantitative evidence evaluating interventions to promote adherence to medications, which included antihypertensives 5 were

of 'high' overall quality,[54 73 81 98 103] 3 were 'moderate' quality,[53 65 95] 8 were of 'critically low' quality,[34 44 46 48 57 63 67 96] and 33 were of critically low quality.[35 37 39–41 43 45 46 49 51 56 62 66 68–71 74–76 78 79 81 86 87 89–94 101 102]

Four 'critically low' quality synthesised quantitative evidence to optimise the prescribing of medications, including statins and antihypertensives.[85 87 90 106] One 'high' quality review evaluated interventions to optimise adherence to both groups of medication.[98 99] Two 'critically low' quality reviews[38 71] and one 'high' quality review synthesised evidence evaluating interventions to optimise the prescribing of antihypertensives alone.[72]

### Discontinuing treatment

Three reviews, one of 'low' quality,[82] five of 'critically low' quality[36 47 88 97 105] synthesised quantitative evidence evaluating interventions promote deprescribing.

One systematic review of quantitative evidence evaluated impact of Medicare Part D on use of under and overuse of medication. Statins and antihypertensives were only two types of medication included in this review.[102]

One review of 'low' quality focused on identifying potentially inappropriate prescribing in older adults with dementia.[26]

Two 'critically low' quality reviews of qualitative evidence; one lay perspectives on hypertension and drug adherence,[55] one reported on patient and/or carer views barriers to and enablers of deprescribing where antihypertensives were the medication of interest.[61]

Two 'critically low' quality reviews synthesised quantitative evidence to optimise the prescribing of antihypertensives.[38 52]

### Prescribing guidelines

One 'critically low' quality,[106] one 'moderate' quality[82] and one 'high' quality review synthesised quantitative evidence evaluating interventions to improve adherence to CVD prescription guidelines.[72]

### DISCUSSION

We aimed to provide a resource to enable policy-makers, commissioners, clinicians and other decision-makers to find and view existing systematic review evidence relevant to optimising the prescribing of statin and antihypertensive medication. The EGM displays systematic review evidence synthesising quantitative evidence evaluating interventions to improve or optimise medication adherence or prescribing and also qualitative evidence examining the perspectives of patients, carers and clinicians of issues which can affect optimal prescribing. This evidence is mapped onto a patient care pathway, according to the type of medication of interest, enabling map users to find and view the evidence most suited to their needs.

The map highlights a cluster of systematic review evidence evaluating interventions to promote adherence to medication, particularly antihypertensive, relevant to the 'pretreatment' and 'maintaining treatment' sections

of the patient care pathway. An identified gap within the pathway is that of qualitative evidence which seeks to explore the views and/or experiences of these types of intervention.

## Strengths and limitations

This EGM will be a useful resource for decision makers who are seeking systematic review evidence to answer questions relating to the optimal prescribing of statins or antihypertensive medication. The map functionality means that the key characteristics of the evidence are easily displayed and allow map users to access a summary or link to full texts as required. The map also identifies both clusters of existing systematic review evidence and gaps, highlighting where further systematic review evidence may or may not be required. However, this map includes only systematic review evidence and thus there may be primary research relevant to the topic which may sit within the identified gaps. In addition, the majority of evidence in this map was of critically low quality. These systematic reviews may be of more limited use to decision-makers and mask gaps where further evidence syntheses or commissioning of primary research is required.

Our search strategy did not capture systematic reviews used to inform NICE guidelines relevant to the review topic. These reviews were not retrieved by our bibliographic database searches, highlighting issues regarding the visibility of the reviews contributing to NICE guidelines. It was beyond our resources to search through individual guidelines to locate relevant reviews.

## Implications of our findings

For research commissioners, the map highlights where further research may be useful, including the following specific areas:

► Further synthesis of existing systematic reviews of experiences of taking or adhering to statin and/or antihypertensive medication.
► Searching for primary studies where gaps in high-quality systematic review evidence have been identified, for example, qualitative evidence regarding patient and/or prescriber experiences of specific interventions to increase medication adherence or practitioner experiences of prescribing statin medication.

The clusters of existing research may also be useful in informing government policy, however, caution is required as many of the systematic reviews are of poor quality which may limit the confidence that can be placed in their findings.

For clinicians, the qualitative evidence within the map highlights factors which could influence patient's seeking out or accessing health services for CVD and their willingness or ability to take any medication which is prescribed to them. The map also includes systematic review evidence evaluating interventions to optimise medication adherence, which may also be useful to inform clinical practice. Patients may find the EGM a useful resource to enable them to review the research associated with the care they receive.

## CONCLUSIONS

This EGM presents a snapshot of the systematic review evidence available to inform the optimal prescribing of statin and antihypertensive medication, displayed to indicate the quality and quantity of evidence available to inform decision-making at key points in the patient care pathway. The evidence included within the map may support policy-makers and service commissioners or aid in the commissioning of further research. The map may also help inform the prescribing decisions of clinicians. While some caution is required when interpreting the content of the map given the limitations of the included evidence and methodology used within this mapping review as indicated above, this map provides a useful resource for a variety of stakeholders to access recent systematic review evidence in this topic area.

**Author affiliations**
[1]Exeter Policy Research Programme Evidence Review Facility, Faculty of Health and Life Sciences, St Luke's Campus, University of Exeter, EX1 2LU, Exeter, UK
[2]NIHR ARC South West Peninsula Patient and Public Engagement Group, University of Exeter, Exeter, UK
[3]European Centre for Environment and Health, University of Exeter, Exeter, UK

**Acknowledgements** Sue Whiffin and Jenny Lowe for administrative support. Morwenna Rodgers for support developing the protocol and search strategy. Kirstin Liabo for support developing the protocol and facilitating involvement of patients and members of the public with the review. The PenARC Patient Engagement Group for their insight throughout the review process. Dr Phil Evans for his support developing the patient care pathways and feedback on the draft evidence and gap map. Melissa Bond from the EPPI centre for continued support in developing the evidence and gap map.

**Contributors** LS, lead author: led scoping process and contributed to the development of the protocol and conduct of screening, data extraction, quality appraisal, led development of the evidence-and-gap map, patient and public involvement and write up of paper and supported liaison with government and clinical stakeholders. SB, information specialist and reviewer: supported scoping and led to the development of the search strategy to inform the protocol, contributed to screening, data extraction and quality appraisal and was responsible for data management throughout the project, developed PRISMA diagram and contributed to all stakeholder involvement and write up of paper. MPN, reviewer: supported development of the protocol and contributed to screening, data extraction, quality appraisal and write up of paper. HML, reviewer: supported screening, data extraction and quality appraisal and reviewed final paper. GJM-T: supported conceptualisation of the project and development of the protocol, supported development of the evidence and gap map and government stakeholder involvement, read and reviewed final paper. MT: patient and public involvement representative and lead, supported development of the protocol and provided feedback on main findings and read and reviewed final paper. RG: supported conceptualisation of the project and development of the protocol, supported development of the evidence and gap map and government stakeholder involvement and read and reviewed final paper. JTC, guarantor of work: supported conceptualisation of the project and development of the protocol, supported development of the evidence and gap map and led government stakeholder involvement and read and reviewed final paper.

**Funding** This paper represents part of a larger piece of work commissioned by the National Institute of Health Research Policy Research Programme by the Exeter PRP Evidence Review Facility (NIHR200695) following a request to the Department of Health and Social Care (DHSC) Research and Development Committee by NHS England and Improvement (NHSE-I). This paper is independent research funded by the National Institute for Health and Care Research Policy Research Programme

(NIHR200605) and supported by the National Institute for Health and Care Research Applied Research Collaboration South West Peninsula.

**Disclaimer** The views expressed in this publication are those of the author(s) and not necessarily those of the National Institute for Health and Care Research, the Department of Health and Social Care or NHS England and Improvement.

**Competing interests** None declared.

**Patient and public involvement** Patients and/or the public were involved in the design, or conduct, or reporting, or dissemination plans of this research. Refer to the Methods section for further details.

**Patient consent for publication** Not applicable.

**Provenance and peer review** Not commissioned; externally peer reviewed.

**Data availability statement** Data are available on reasonable request. For access to documentation supporting this review, please contact LS.

**ORCID iDs**
Liz Shaw http://orcid.org/0000-0002-6092-5019
Hassanat Mojirola Lawal http://orcid.org/0000-0001-5996-5139
Jo Thompson Coon http://orcid.org/0000-0002-5161-0234

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
