## [Reviewer comments · BMJ Open]

ARTICLE DETAILS

TITLE (PROVISIONAL)	What is the quantity, quality and type of systematic review evidence available to inform the optimal prescribing of statins and antihypertensives? A systematic umbrella review and evidence and gap map
AUTHORS	Shaw, Liz; Briscoe, Simon; Nunns, Michael; Lawal, Hassanat; Melendez-Torres, G.J; Turner, Malcolm; Garside, Ruth; Thompson Coon, Jo

VERSION 1 – REVIEW

REVIEWER	Chih-Lin Chi University of Minnesota, School of Nursing & Institute for Health Informatics
REVIEW RETURNED	04-Apr-2023

GENERAL COMMENTS	The manuscript aims to map the systematic evidence to inform the optimal prescribing of statins and antihypertensives. They particularly focus on the creation of evidence and gap map to display the quantity, quality, and type of quantitative systematic evidence. As the authors indicate that diverse range of studies supports the need of those living with or at risk of CVD. There are a few comments to improve the manuscript. (1) How to determine 2010 as a cut point for the systematic reviews? (2) The method and result sections are not clear. A workflow that indicate what they are doing can help? Currently, page 28 seems to serve that function, but number and labels do not match. As a result, it is difficult to understand the flow. (3) Quality of Fig 1 and 3 need to improve. (4) Where is Fig 2? Is it in P.28? (5) It is hard to match Table 2 and content to Fig 1,2,3. Thus, hard to match contents with Table and Figures. (6) Again, it is not intuitive to follow P.12 to Figure 3. In lines 48-55, the authors can indicate the colors of clusters.
---

REVIEWER	Sara Malo Universidad de Zaragoza
REVIEW RETURNED	03-May-2023

GENERAL COMMENTS	Original and useful study that provides a method to collect information of interest for both policy makers and clinicians. Some aspects that should be addressed to improved the quality and understanding of the manuscript are detailed below: - In the title, the expression "What is the quantity, quality and type of..", should be rewritten, e.g. "Is systematic review evidence available to inform..." or "Does systematic review evidence exist...?". Moreover, your study is focused on both prescribing and
---

	adherence patterns, so both terms should be mentioned in the title and the objective, not only "prescribing". Finally, you mention "medication to treat cardiovascular disease", but detailing the "statins and antihypertensive drugs" is preferable, given that cardiovascular preventive therapy include more therapeutic agents. You should justify why you consider so many databases to search the literature on the study topic. IN the Strenghts and limitations section, the third point is not completely clear. please, clarify it "...highlighting issues regarding...".  - The Introduction section is mainly focused on data and practices in UK. It should be more opened to other contexts. - IN table 1, the outcome defined should be more extensively defined. - The three phases conforming the adherence process should be denominated as the ABC taxonomy on mediation adherence proposes, i.e. Initiation, implementation (not maintaining) and discontinuation. - I miss a more detailed description/reflection in the Discussion on how to transfer the findings to the actors implied. PLease, include it.
--	--

VERSION 1 – AUTHOR RESPONSE

Reviewer 1	How to determine 2010 as a cut point for the systematic reviews?	Please see our response to the editors comment above.
Reviewer 1	The method and result sections are not clear. A workflow that indicate what they are doing can help? Currently, page 28 seems to serve that function, but number and labels do not match. As a result, it is difficult to understand the flow.	Thank you for your comment. We have provided additional signposting throughout our methods section to support readers who may be unfamiliar with the process of conducting systematic reviews. Our results section first provides an overview of all 80 reviews included in our umbrella review. It then focuses on describing the characteristics of the 68 reviews included in the Evidence and Gap map (a rationale for this separation is provided at the start of the results section). To clarify this, we have added subheadings and amended the start of the paragraph at the start of each of these sections to clarify the number of papers we are referring to; and whether these are in relation to all included studies, or just those included in the EGM.
Reviewer 1	(3) Quality of Fig 1 and 3 need to improve.	Thank you for your comment, we have revised these figures.
Reviewer 1	(4) Where is Fig 2? Is it in P.28?	We are not sure of the format of the document you were provided with. We apologise for the confusion.
Reviewer 1	(5) It is hard to match Table 2 and content to Fig 1,2,3. Thus, hard to match contents with Table and	The table and figures were intended to provide alternative ways of representing the information according to the preferences and intended purpose of the reader. We agree that it would indeed be very

	Figures.	difficult and time consuming to cross-reference the content of all of these. Instead, we expect that readers of the paper would access the map to explore it's content themselves.
Reviewer 1	(6) Again, it is not intuitive to follow P.12 to Figure 3. In lines 48-55, the authors can indicate the colors of clusters.	Thank you for your comment. Again, we do not have access to the document with which you were provided so it is unclear exactly which lines you were pertaining to. However, we hope the additional signposting (described above) have improved the clarity of our reporting. As indicated above, we would expect readers of the paper to open the evidence and gap map to explore it's content themselves, rather than rely on the content of the paper alone.
Reviewer 2	In the title, the expression "What is the quantity, quality and type of..", should be rewritten, e.g. "Is systematic review evidence available to inform...?" or "Does systematic review evidence exist...?. Moreover, your study is focused on both prescribing and adherence patterns, so both terms should be mentioned in the title and the objective, not only "prescribing". Finally, you mention "medication to treat cardiovascular disease", but detailing the "statins and antihypertensive drugs" is preferable, given that cardiovascular preventive therapy include more therapeutic agents.	The title structure is standardised and typical of that used within umbrella reviews of this nature. It reflects the aim/research question of this review (as per journal guidelines), which was not to establish whether there was evidence (of which our scoping had already shown there was an abundance), but what the nature of this evidence was. We recognise that 'optimal prescribing' occurs as a result of an interaction between the behaviour of the prescribing clinicians and the behaviour of the person to whom the medication is being prescribed, and that the combined behaviour of this dyad occurs within broader service/familial/government policy contexts. Thus, we used the term 'optimal prescribing' to encompass these factors, which is explained towards the end of the background section of this paper and reflected in the review inclusion criteria. We have amended the sentence within the "objectives" section of our abstract to read "...statin and antihypertensive medication" as requested.
Reviewer 2	You should justify why you consider so many databases to search the literature on the study topic.	The number and selection of databases is typical for an umbrella review and reflects our desire to comprehensively search for systematic review evidence within this field and the broad scope of an umbrella review. The databases selected provide coverage of a variety of health care disciplines which are relevant to our research question. We do not

		think this needs justifying within the paper.
Reviewer 2	IN the Strengths and limitations section, the third point is not completely clear. please, clarify it "...highlighting issues regarding...".	We have reworded this point to read: "These reviews were not retrieved by our bibliographic database searches, thus highlighting issues regarding the lack of visibility of the reviews contributing to NICE guidelines."
Reviewer 2	The Introduction section is mainly focused on data and practices in UK. It should be more opened to other contexts.	Thank you for your comment. Our research was commissioned by the UK Department of Health and Social Care, which is explicitly stated and reflected in the UK policy focus in the background of this paper. We have amended the background section to provide more global context for this review
Reviewer 2	IN table 1, the outcome defined should be more extensively defined.	Thank you for your comment. We have amended this segment of the table to read: "Relevant to the aims of our review and stated clearly in the study abstract. For reviews of quantitative evidence, examples may include:  - Measures of patient adherence; - Measures of prescriber adherence to prescribing guidelines; - Measures of success of implementing interventions to promote adherence or prescribing. For reviews of qualitative evidence, please see phenomenon of interest section above."
Reviewer 2	The three phases conforming the adherence process should be denominated as the ABC taxonomy on mediation adherence proposes, i.e. Initiation, implementation (not maintaining) and discontinuation.	The structure of our evidence and gap map was discussed and approved by patients, health care professionals and national policy makers. Therefore, whilst there will clearly be other potential ways to label elements of the pathway and to organise the evidence, we do not believe it would be appropriate to make the suggested changes on the basis of a single viewpoint. As the proposed taxonomy already aligns closely with what we have used and therefore would not add significant additional value to the presentation or interpretation of findings, we have not made this change.
Reviewer 2	I miss a more detailed description/reflection in the Discussion on how to transfer the findings to the actors implied. Please, include it.	We are a little unsure about what the reviewer would like to see. As described in the discussion, the map is an interactive resource which will enable map users to access and explore the information relevant to their needs. This may include patients, health care professionals, local or national policy makers using

		the evidence to inform decision making.
--	--	---

VERSION 2 – REVIEW

REVIEWER	Sara Malo Universidad de Zaragoza
REVIEW RETURNED	11-Jul-2023

GENERAL COMMENTS	I would like to thank the authors for the efforts in the application of the comments/modifications proposed. Most of them have been addressed. However, there are still three issues that should be amended:  - Prescribing is referred to the act during which the doctor makes a diagnosis and prescribes a drug. It is then up to the patient to decide whether and how to take it. The first act is called prescription. The second is adherence to the medication. Therefore, "optimal prescribing" is not the interaction between the behaviour of the clinician and the behaviour of the person to whom the medication is being prescribed. Both terms should be mentioned as requested. - Although the selection of such a large number of databases is typical for an umbrella review, the authors can explain the reasons in the text ("desire to comprehensively search for systematic review evidence..."). - Finally, regarding the comment "I miss a more detailed description/reflection in the Discussion on how to transfer the findings to the actors implied. Please, include it.", I would like the authors to explain a bit more in detail the usefulness and applicability of the tool, in particular for patients, who are not mentioned.
--

VERSION 2 – AUTHOR RESPONSE

Reviewer comment	Author response
I would like to thank the authors for the efforts in the application of the comments/modifications proposed. Most of them have been addressed. However, there are still three issues that should be amended:  - Prescribing is referred to the act during which the doctor makes a diagnosis and prescribes a drug. It is then up to the patient to decide whether and how to take it. The first act is called prescription. The second is adherence to the medication. Therefore, "optimal prescribing" is not the interaction between the behaviour of the clinician and the behaviour of the person to whom the medication is being prescribed. Both terms should be mentioned as requested. 	Thank you for your thoughts. I think this is an issue on which we appear to disagree and we refer this to the editor for their opinion. The authors strongly feel that the prescribing behaviour of a clinician does not (or should not) occur in a vacuum and is not a "one way" didactic style of interaction which is completely independent from the behaviour of the patient. Instead, we see prescribing as a process which often involves conversation and shared decision making between clinician and the patient, influenced by the patient's individual circumstances, across multiple timepoints. We remain of the opinion that the definition of "optimal prescribing" used in this study more accurately portrays this interaction, whilst also recognising the separate behaviours of "prescribing" and "adherence" which are encompassed by our original definition.

	This definition was approved by our stakeholders, including prescribing practitioners and government commissioners, and to change it at this stage would undermine the transparency and replicability of the review process.
- Although the selection of such a large number of databases is typical for an umbrella review, the authors can explain the reasons in the text ("desire to comprehensively search for systematic review evidence...").	We have added the following to the search strategy section: "This selection of databases reflects our desire to comprehensively search for systematic review evidence within this field and the broad scope of an umbrella review. The databases selected provide coverage of a variety of health care disciplines relevant to our research question." We have also added the following justification for our date limit for searches "to prioritise the most recent systematic reviews, evaluating interventions and experiences most relevant to current medical settings, for inclusion in the EGM."
- Finally, reagrding the comment "I miss a more detailed description/reflection in the Discussion on how to transfer the findings to the actors implied. PLease, include it.", I would like the authors to explain a bit more in detail the usefulness and applicability of the tool, in particular for patients, who are not mentioned.	Thank you for your thoughts. We have added the following to the end of the 'Implications' section: "Patients may find the evidence and gap map a useful resource to enable them to review the research associated with the care they receive."